# Detection of the Omicron SARS-CoV-2 Lineage and Its BA.1 Variant with Multiplex RT-qPCR

**DOI:** 10.3390/ijms232416153

**Published:** 2022-12-18

**Authors:** Nikita D. Yolshin, Andrey B. Komissarov, Kirill V. Varchenko, Tamila D. Musaeva, Artem V. Fadeev, Dmitry A. Lioznov

**Affiliations:** 1Smorodintsev Research Institute of Influenza, 197376 Saint Petersburg, Russia; 2Department of Infectious Diseases and Epidemiology, First Pavlov State Medical University, 197022 Saint Petersburg, Russia

**Keywords:** SARS-CoV-2 RT-qPCR, SARS-CoV-2 variants detection, Omicron RT-qPCR, Omicron detection, BA* qPCR, BA.1 qPCR, Ins214EPE, ERS31del, SARS-CoV-2 indels

## Abstract

Whole genome sequencing (WGS) is considered the best instrument to track both virus evolution and the spread of new, emerging variants. However, WGS still does not allow the analysis of as many samples as qPCR does. Epidemiological and clinical research needs to develop advanced qPCR methods to identify emerging variants of SARS-CoV-2 while collecting data on their spreading in a faster and cheaper way, which is critical for introducing public health measures. This study aimed at designing a one-step RT-qPCR assay for multiplex detection of the Omicron lineage and providing additional data on its subvariants in clinical samples. The RT-qPCR assay demonstrated high sensitivity and specificity on multiple SARS-CoV-2 variants and was cross-validated by WGS.

## 1. Introduction

The global pandemic caused by severe acute respiratory syndrome coronavirus 2 (SARS-CoV-2) started in 2019. More than 640 million people have been infected and have suffered from the coronavirus disease 2019 (COVID-19) to date (November 2022) [1]. The virus, which killed more than 6,600,000 people in two years, continues to evolve and spread throughout the globe [2] despite various non-pharmaceutical interventions and the availability of plenty of vaccines.

Several months after SARS-CoV-2 appeared, new distinct lineages were detected. One after the other, lineages emerged and replaced previous variants. Practically worldwide, the “wave” of the Delta variant put an end to the Alpha variant dominance, only to be replaced by Omicron BA.1 in December 2021–January 2022. However, BA.1 domination did not persist and was replaced by BA.2 in March 2022. New variants causing “epidemic waves” often demonstrate higher transmission rates and decreased susceptibility to neutralization by pre-existing antibodies against SARS-CoV-2 due to previous infection or vaccination [3]. Omicron (B.1.1.529) was designated a variant of concern (VOC) by WHO on 26 November 2021. Omicron shares some mutations with the Delta variant, including D614G and T478K, which facilitate ACE2 binding and fusion with human cells [4,5]. Omicron has also acquired new mutations that contribute to immune evasion by masking highly immunogenic sites. Thus, the Omicron variant poses an increased risk of reinfection with SARS-CoV-2 as compared to other VOCs because it is able to resist neutralization by antibodies [6,7] or sera from vaccinated people [8]. The Omicron lineage includes several subvariants with strikingly different genetic characteristics. The most widespread subvariants are BA.1 and BA.2, with more than 2.3 and 1.1 million genomes, respectively, shared via the Global Initiative on Sharing All Influenza Data (GISAID) [9] by May 2022.

Quantitative (q) PCR plays a key role in the rapid and easily scalable assessment of SARS-CoV-2 variants’ spread, e.g., in outbreak scenarios. Although next-generation sequencing (NGS) is performed routinely for surveillance purposes, it is limited in sample number and, hence, less informative. It is of special concern in the case of low- and middle-income countries and other limited resource settings where sequencing capacity is low [10]. In the first assessment of the Alpha variant (B.1.1.7), population-wide frequency was performed by analyzing so-called *S* gene target failure (SGTF) in widely used commercial qPCR assays; false negatives for the *S* gene target occasionally proved to be a good proxy estimate of the B.1.1.7 spread. SGTF was found to be a reliable marker of Alpha after validation with whole-genome sequencing (WGS) [11]. SGTF was also used for Omicron detection [12]. However, it is not a specific test since it identifies other lineages carrying the 69/70 deletion and some Omicron variants do not carry this mutation.

qPCR assays for tracking the emerging variants of SARS-CoV-2 make surveillance cheaper and faster, which is critical for promptly implementing public health measures. qPCR relies on detecting unique mutations as a sign of emerging variants. The surveillance of SARS-CoV-2 evolution has been focused on the detection of nucleotide substitutions. qPCR assays could be targeted at single nucleotide polymorphisms (SNP), but this application would require highly specific enzymes and chemistry that would be too expensive for epidemiological studies. SNPs can be detected by high-resolution melting (HRM) technology. However, the high mutability of viral genomes makes HRM too sensitive to emerging mutations, thus making it hard to interpret [13].

Insertion-deletion polymorphisms (indels) would make a more feasible qPCR target, allowing for less laborious assay development and optimization. Increased numbers of indels seem to mediate immune escape and growing population resistance due to vaccination and previous infections. Indels occur most frequently in the Spike protein but are also found in others, especially those involved in interactions with the host immune system [14]. The Omicron lineage (BA*) was found to have a unique deletion that proved to be a good target for its detection, while the BA.1 variant sequence also contains unique lineage-specific insertion.

Here, we report on a novel multiplex RT-qPCR test which includes (1) a generic assay detecting all known SARS-CoV-2 variants, (2) an assay targeting all subvariants of the Omicron lineage, (3) an assay specific to BA.1, labeling the non-BA.1 variants as drop-outs, and (4) internal control reaction. The multiplex RT-qPCR assay simultaneously detects any SARS-CoV-2 and discriminates the Omicron lineage and its BA.1 SARS-CoV-2 subvariant in RNA samples in a single reaction, thus simplifying testing and reducing test costs.

## 2. Results

### 2.1. Primer and Probe Design

Oligonucleotide primers and probes targeting the conservative region in *ORF1* (Figure 1) were designed to detect any SARS-CoV-2 RNA and were used in this multiplex assay as an indicator of SARS-CoV-2 in a clinical sample (2-SARS-CoV-2-ORF-1). The primer and probe sequences were confirmed to have perfect matches within all SARS-CoV-2 genome sequences available from GISAID to the moment.

A unique ins214EPE insertion is located in the area termed an “insertion hotspot” in the Spike protein-encoding gene of BA.1 [15]. Hence, this mutation was incorporated in the RT-qPCR design to detect the Omicron BA.1 lineage (Figure 2). Long insertions are thought to be generated via the template-switching mechanism associated with the synthesis of subgenomic RNAs, while short inserts seem to result from RdRP slippage on short runs of A or U [16].

All Omicron lineages (BA*) carry a three amino acid deletion ERS31del in the N gene encoding the Nucleocapsid protein (Figure 3). According to metadata, this mutation is found in 93% of BA* genomes submitted to the GISAID EpiCoV repository by the end of April 2022. Therefore, it was used to design a qPCR assay discriminating Omicron from other SARS-CoV-2 variants (Figure 4). ERS31del is considered lineage-defining and is found in all BA* genomes. However, it is not annotated in all the genomes, presumably because algorithms struggle with processing NGS data if it contains insertions and deletions since accurate indel calling is known to be an issue [17]. Assays were designed to amplify DNA from RNA templates derived from different SARS-CoV-2 lineages while detecting insertions and deletions with the probes complementary to a sequence containing the corresponding variant-specific mutation (Figure 3 and Figure 4).

### 2.2. Sensitivity, Specificity, and Amplification Efficiency Assessment

The limit of detection (LOD) was determined by serial dilution of full-length SARS-CoV-2 RNA and probit regression analysis (Table 1, Figure 5). Standard SARS-CoV-2 RNA concentration was calculated with digital (d) PCR (QIAcuity Digital PCR System) using SARS-CoV-2 detection kit (ModularDx, TIB Molbiol). The concentration amounted to 49,860 copies per microliter. All primer-probe sets detected SARS-CoV-2 at 1000 virus copies per ml at the lowest.

The developed multiplex RT-qPCR assay demonstrates high specificity. It was preliminarily tested on 28 clinical samples of RNA extracted from oropharyngeal swabs, which contained 11 SARS-CoV-2 lineages according to WGS data (Table 2). Specific signal was detected only in samples with SARS-CoV-2 Omicron lineage RNA.

The specificity of the multiplex RT-qPCR assay was further tested on another 96 samples containing variants previously defined by WGS: 32 non-BA.1 Omicron, 32 BA.1, 22 non-Omicron variants, and 10 negative controls. The experiment yielded 100% true positives and 100% true negatives (Table 3). Detailed data is provided in Appendix A, Table A1.

Ten-fold serial dilutions of Omicron RNA were used to assess PCR efficiency. The PCR efficiency of 2-SARS-CoV-2, Ins214EPE, and ERS31del assays reached 108%, 98.9%, and 105%, respectively. The graphs are presented in Appendix B (Figure A1, Figure A2 and Figure A3).

Additionally, the specificity was tested on clinical samples that tested positive for other respiratory viruses from the collection of Smorodintsev Research Institute of Influenza, such as influenza and parainfluenza viruses, human seasonal coronaviruses (OC43, NL63, 229E, HKU1), hRSV, rhinoviruses, bocaviruses, and metapneumovirus (33 in total). The test yielded no false-positive results. Data is shown in Appendix C (Table A2).

After combining the four reactions into a single multiplex, its accuracy and specificity were found to be similar to those of a single RT-qPCR (Appendix D, Table A3). Thus, the developed multiplex RT-qPCR assay provides diagnostic performance comparable to currently used singleplex RT-qPCR tests while requiring fewer reagents and less time. Interpretation of the RT-qPCR results is presented in Table 4.

### 2.3. Implementing the Developed Assays in SARS-CoV-2 Surveillance in Russia

More than 30,000 specimens were tested with the multiplex RT-qPCR assay while screening clinical specimens collected in various Russian Federation regions from December 2021 to May 2022. The multiplex RT-qPCR allowed us to monitor Omicron variant frequencies from its first appearance and initial spread to the highest epidemic peaks since the beginning of the COVID-19 pandemic in Russia (Figure 6).

Thus, we developed a sensitive and specific multiplex RT-qPCR, which detects the SARS-CoV-2 Omicron lineage and specifically identifies the BA.1 subvariant, and verified it with WGS on plenty of clinical samples.

The Ins214EPE assay protocol was shared with the scientific community on protocols.io in December 2021 [18]. Afterward, the multiplex assay was developed and the protocol was also made available on protocols.io in February 2022 [19]. The described RT-qPCR assay is patented in Russia (#2779025) and commercialized by the BioLabMix company.

## 3. Discussion

Epidemiological and clinical research requires collecting data on the spreading of emerging SARS-CoV-2 variants. Some emerging mutations are good targets for detecting particular viral genomes. RT-qPCR can detect SNPs but demands highly specific enzymes and chemistry, such as PACE-RT chemistry [20]. HRM technology is a suitable technique to detect SNPs in some applications. However, its robustness is jeopardized if the target is genetically diverse or mutable [21]: the presence of more than one mutation site can result in complex melt curves which may be difficult to interpret. Hence, HRM is not perfect in the case of the *S* and *N* genes of SARS-CoV-2.

SGTF assay [11], the so-called “drop-out” technique, also has an inherent disadvantage in this kind of test. It may fail to detect the *S* target in swab samples with lower virus concentration, resulting in false negatives.

Other RT-qPCR assays for Omicron detection are known. One of the assays identifies the Omicron lineage [22] based on oligonucleotide complementarity to eight nucleotide polymorphisms in the SARS-CoV-2 Omicron genome. The assay uses the SNP cluster in the *S* gene as a marker of the Omicron lineage. However, some of these point mutations, such as E484Q and N501Y, are present in non-Omicron SARS-CoV-2 variants as well, while others are absent in some Omicron subvariants, such as Q493R in BA.4&5. The main limitation of this method is the necessity to run another SARS-CoV-2 assay in parallel for Cq comparing since there is also a weak signal for the Wuhan Hu-1, the Alpha, Beta, Delta, and Gamma variants. The authors also noticed that the temperature profile of amplification is critical to assay specificity. It could also hinder the reproducibility of this test if different RT-qPCR reagents are used. Another described Omicron detection technique relies on Sanger sequencing following amplification [23]. However, this method is laborious, expensive, difficult to scale up, and thus could hardly be used for screening.

Nevertheless, SARS-CoV-2 emerging variants often carry not only SNPs but also lineage-defining indels. Targeting these with PCR could be a less laborious approach to develop and optimize. Some published assays for Omicron lineage detection are based on identified 69/70 deletion [24,25], but widespread BA.2 subvariants of Omicron lack this mutation. The assay developed in our laboratory avoids the abovementioned disadvantages and carries only limitations specific to the qPCR method, such as sensitivity to virus genetic variability at the points of oligonucleotide annealing.

Two indels unique to the Omicron variant, Ins214EPE and ERS31del, were chosen for RT-qPCR assays designed to detect the Omicron lineage and its BA.1 subvariant. The developed multiplex assay for Omicron and its BA.1 variant detection has been in use for more than 9 months. Despite the rapid virus evolution, ERS31del in the N gene is still present in all circulating variants, including new immune-evasive variants with global spread BQ and XBB, and Ins214EPE is still a unique marker of the BA.1 variant to the moment of article submission (November 2022). The assay is cheap, fast, robust, and simple to be introduced and implemented in any PCR laboratory.

The developed assays, including the multiplex RT-qPCR, demonstrate high specificity and sensitivity. The assays were verified, broadly tested, and used to screen thousands of samples. The final multiplex platform included the ERS31del assay for the Omicron lineage (BA*) detection, Ins214EPE assay against the BA.1 variant, ORF1 SARS-CoV-2 for the detection of any type of SARS-CoV-2, and the RP assay for human RNA amplification as a control for RNA extraction, sample storage, and PCR conditions. For kits based on these assays, RNA controls were also obtained.

ERS31del is typical for all Omicron subvariants detected by July 2022: BA.1, BA.2, BA.3, BA.4, BA.5, BA.2.12.1, and BA.2.75. Thus, the ERS31del assay can detect all the variants that are spreading in 2022. However, some SNP-specific assays are needed to discriminate Omicron subvariants since no specific indels are found (except for BA.1). Similarly to SGTF, which has already been proved useful twice during the pandemic, the Ins214EPE and ERS31del assays could be useful for targeting novel SARS-CoV-2 variants, too. One or several SARS-CoV-2 variants are highly likely to become a future seasonal infection; thus, the new RT-qPCR assays are a promising approach to seasonal SARS-CoV-2 identification.

## 4. Materials and Methods

### 4.1. Sample Collection and RT-PCR Testing

De-identified samples used in this study were collected during the ongoing surveillance of SARS-CoV-2 variability routinely conducted by the Smorodintsev Research Institute of Influenza under the Coronavirus Russian Genetic Initiative (CORGI). Written informed consent was obtained from all subjects following the order of the Ministry of Health of the Russian Federation of 21 July 2015 #474 n. This study was reviewed and deemed exempt by the Local Ethics Committee of the Smorodintsev Research Institute of Influenza (protocol No. 152, 18 June 2020).

Nasopharyngeal and throat swabs were collected in virus transport media. Total RNA was extracted using the Auto-Pure 96 Nucleic Acid Purification System (Allsheng, Hangzhou, China) and the NAmagp DNA/ RNA extraction kit (Biolabmix, Novosibirsk, Russia). Extracted RNA was immediately tested for SARS-CoV-2 using Biolabmix SARS-CoV-2 RT-PCR Detection System (Biolabmix, Novosibirsk, Russia), following the Hong Kong University protocol with modifications [26].

To assess the specificity of the assays, a panel of respiratory viruses was obtained by routine testing of nasopharyngeal swabs from patients with respiratory infection diseases using PCR assay “ARVI-screen” (Amplisens, Moscow, Russia).

### 4.2. RT-qPCR Assay Design

In silico specificity tests were carried out using NCBI BLAST. Target regions of oligonucleotide sets were searched against the nucleotide sequence database with excluded SARS-CoV-2 genomes. Additionally, Ins214EPE and ERS31del oligonucleotides’ complementarity with BA.1 and BA* genomes submitted to the GISAID repository by the end of January was checked in silico, and the oligonucleotides aligned successfully with the majority of the corresponding genomes.

Sequences of oligonucleotides used in the multiplex RT-qPCR assay are presented in Table 5.

The expected amplicon size of ORF1 is 88 bp, S gene–89 bp, N gene–99 bp. Human RNA polymerase gene amplification assay used as an internal control was developed by the Center for Disease Control and Prevention [27]. All primers and probes were designed manually following basic primer design rules [28] and manufactured by ALCOR BIO company (Russia, Saint-Petersburg). The assays were designed to amplify a DNA fragments, including indels, from RNA templates derived from different SARS-CoV-2 variants. The detecting probes were complementary to a sequence containing the corresponding variant-specific mutation. The slope of the standard curves for assays 2SARS-CoV-2-ORF-1, Ins214EPE, and ERS31del was −3.14, −3.34 and −3.20, respectively. The amplification efficiency exceeded 95% for all primer–probe sets (Appendix B).

Oligonucleotides for the multiplex RT-qPCR were premixed as shown in Table 6.

PCR master mix contained 12.5 μL of PCR buffer, 1.2 μL of oligonucleotide premix (Table 6), 1 μL of enzyme mix, and 5.3 μL of water per reaction. Then, 5 μL of RNA extracted from a clinical sample was added. The thermocycler program included reverse transcription for 15 min at 45 °C, initial denaturation for 5 min at 95 °C, and 40 cycles of denaturation (10 s at 95 °C) followed by annealing/elongation (30 s at 62 °C). Bio-Rad CFX96 machine and Biolabmix reagents for PCR (RT-qPCR kit) were used in this work.

### 4.3. Digital PCR

dPCR was run with the QIAcuity One 5-Plex from QIAGEN (Hilden, DE) using QIAcuity One-Step Viral RT-PCR kit with SARS-CoV-2 detection kit (ModularDx, TIB Molbiol). BA.1 pooled sample was used as template RNA (GenBank: OP810428.1). A total of 5 μL template RNA was added to master-mix containing 3 μL One-Step Viral RT-PCR Master Mix, 0.12 μL 100x Multiplex RT mix, 0.5 TIB oligonucleotides mix, and 3.4 μL of water per reaction. dPCR settings were as follows: 40 min for reverse transcription at 50 °C, 2 min at 95 °C for polymerase activation followed by 40 cycles of 5 s at 95 °C for denaturation, and 30 s at 60 °C for annealing and extension.

### 4.4. Genome Assembly and Consensus Correction

WGS data were obtained by Illumina MiSeq and MinIon (Oxford Nanopore Technology, Oxford, UK) and used to confirm PCR results and create the SARS-CoV-2 lineage panel. Libraries for Illumina sequencing were prepared using the Nextera XT library preparation kit (Illumina, San Diego, CA, USA) and then sequenced on a MiSeq instrument (Illumina, San Diego, CA, USA) with a MiSeq Sequence kit v3. FastQC software was used for sequence data quality assessment. Trimmomatic was applied for quality data trimming. Reads were mapped onto reference sequences using BWA. SAMtools-mpileup v1.10.68 [29] was used to produce draft consensus sequences which were then corrected. Libraries for Oxford Nanopore sequencing were prepared using an SQK-LSK109 DNA Ligation Sequence kit (Oxford Nanopore, Oxford, UK). Sequencing was performed using a MinIon instrument (Oxford Nanopore, Oxford, UK) with a R9.4.1 flowcell. Guppy software was used for base-calling and data quality trimming. Reads were mapped onto the reference sequence Wuhan-Hu-1 SARS-CoV-2 using Minimap2 [30]. Lineage was defined using PANGO algorithms.

### 4.5. Molecular Cloning

Positive control RNA for RT-qPCR assays was produced via in vitro transcription (IVT) from pDNA templates, obtained by cloning the corresponding viral cDNA fragments to pJet using CloneJet Kit (Thermo Scientific, Waltham, USA). Bacterial clones were grown in LB media with ampicillin 100 μg/mL and plasmid DNA was extracted with Evrogen (Russia) plasmid midi kit.

## 5. Patents

The following three patents registered in Russian provide the details about the described RT-qPCR assays: Nº2761481, Nº2772362, and Nº2779025 for 2-SARS-CoV-2 test, Ins214EPE test, and multiplex RT-qPCR, respectively.

## Figures and Tables

**Figure 1 ijms-23-16153-f001:**
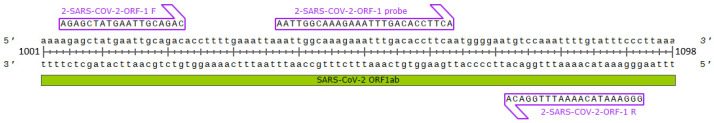
2-SARS-CoV-2-ORF-1 oligonucleotides binding scheme.

**Figure 2 ijms-23-16153-f002:**
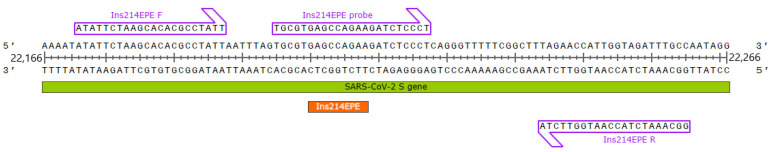
Ins214EPE oligonucleotides binding scheme.

**Figure 3 ijms-23-16153-f003:**
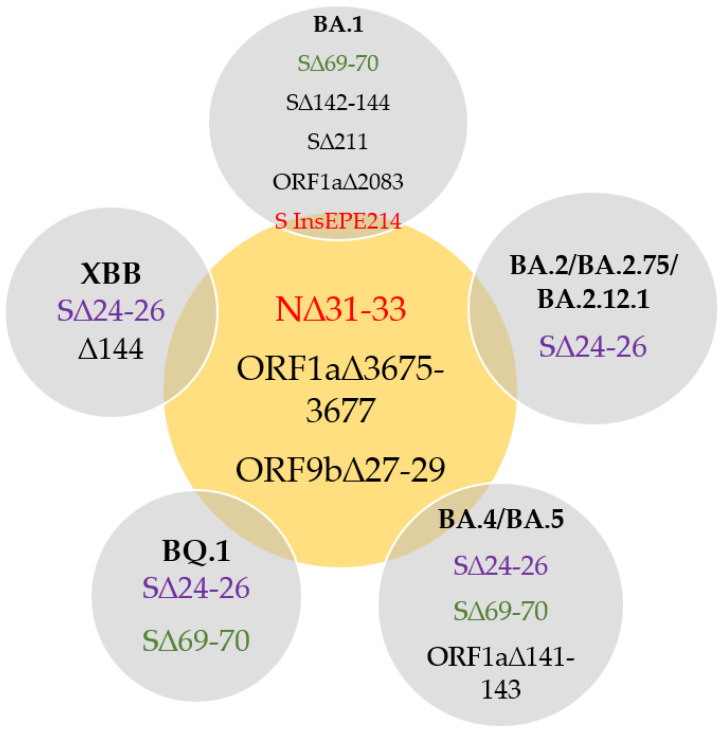
Diagram of the deletions shared by Omicron subvariants.

**Figure 4 ijms-23-16153-f004:**
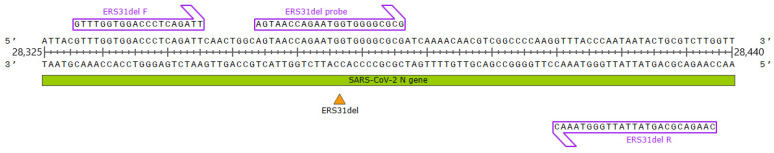
ERS31del oligonucleotides binding scheme.

**Figure 5 ijms-23-16153-f005:**
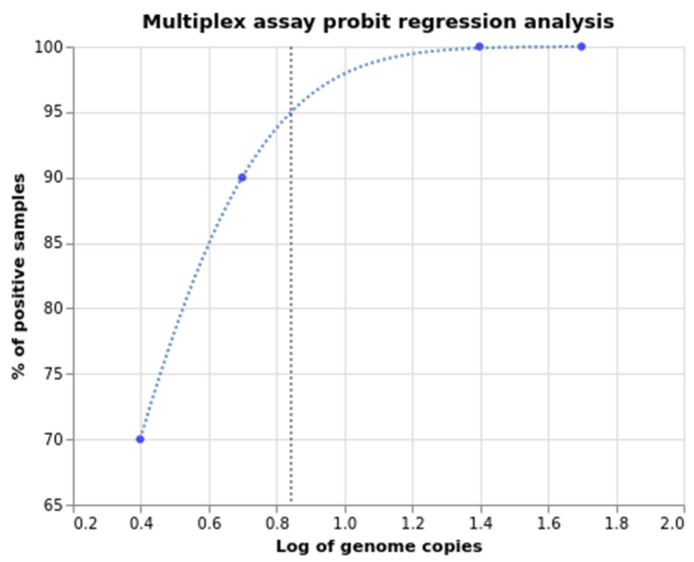
Probit regression model of the multiplex assay LOD.

**Figure 6 ijms-23-16153-f006:**
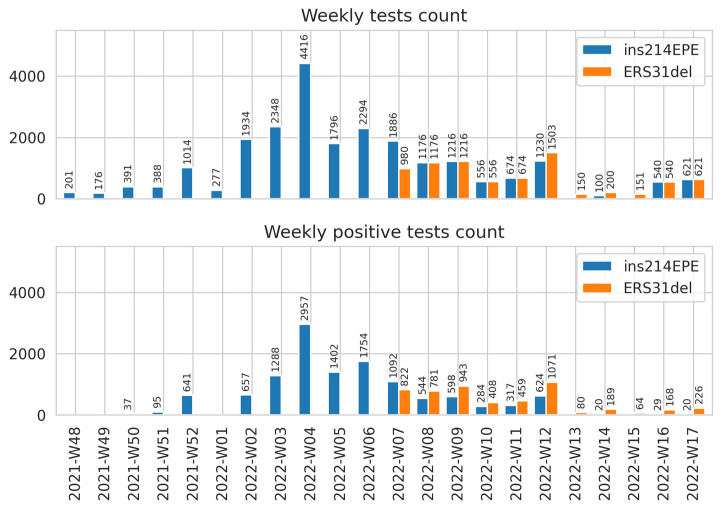
Samples collected in different regions of Russia during Omicron spreading were tested in Smorodintsev Research Institute of Influenza using the developed multiplex RT-qPCR.

**Table 1 ijms-23-16153-t001:** Serial dilution analysis with the multiplex RT-qPCR assay.

Copies per Reaction	Log	% Positives	Signal Ratio
50	1.699	100	20/20
25	1.398	100	20/20
5	0.699	90	18/20
2.5	0.398	70	14/20
LOD (C95) = 0.84 log		LOD (C95) = 7 copies per reaction	

**Table 2 ijms-23-16153-t002:** Assessment of the analytical specificity of the assays to different SARS-CoV-2 variants. “Nd”—“Not detected”.

GISAID #	PANGO Lineage	RP Assay, Cq	SARS-CoV-2 Assay, Cq	ERS31del Assay, Cq	Ins214EPE Assay, Cq
	P.1	35.35	15.45	Nd	Nd
EPI_ISL_1257814	B.1.351	39.91	Nd	Nd	Nd
EPI_ISL_415710	B.1	Nd	15.69	Nd	Nd
EPI_ISL_1652610	B.1	33.84	26.69	Nd	Nd
EPI_ISL_1919527	B.1	24.26	26.98	Nd	Nd
EPI_ISL_2698439	B.1.1.7	25.27	20.64	Nd	Nd
EPI_ISL_2523457	B.1.1.7	30.54	28.05	Nd	Nd
EPI_ISL_2450515	B.1.1.7	26.99	31.25	Nd	Nd
EPI_ISL_3454801	AT.1	30.30	33.07	Nd	Nd
EPI_ISL_3454797	AT.1	30.02	31.33	Nd	Nd
EPI_ISL_3454782	AT.1	28.00	33.00	Nd	Nd
EPI_ISL_2698481	B.1.1.523	29.12	27.09	Nd	Nd
EPI_ISL_2698509	B.1.1.523	27.39	23.46	Nd	Nd
EPI_ISL_2523540	B.1.1.523	30.02	27.09	Nd	Nd
EPI_ISL_2698496	B.1.1.317	25.22	25.31	Nd	Nd
EPI_ISL_2523528	B.1.1.317	27.58	22.71	Nd	Nd
EPI_ISL_2450468	B.1.1.317	28.70	23.91	Nd	Nd
EPI_ISL_4563892	B.1.617.2	26.66	34.03	Nd	Nd
EPI_ISL_5263415	B.1.617.2	25.72	28.43	Nd	Nd
EPI_ISL_4563887	AY.129	26.78	25.42	Nd	Nd
EPI_ISL_6831894	AY.122	23.96	28.95	Nd	Nd
EPI_ISL_6831897	AY.122	31.16	35.05	Nd	Nd
EPI_ISL_6831903	AY.122	24.19	34.93	Nd	Nd
EPI_ISL_11503033	BA.1	27.98	22.20	23.57	22.08
EPI_ISL_10931393	BA.1	27.08	24.93	27.16	24.65
EPI_ISL_10931401	BA.1	25.19	27.48	29.83	25.57
EPI_ISL_9338305	BA.2	26.50	19.64	21.07	Nd
EPI_ISL_9338650	BA.2	24.70	20.81	22.70	Nd

**Table 3 ijms-23-16153-t003:** Multiplex RT-qPCR specificity test on 96 samples panel.

	Omicron BA* Non-BA.1 (32)	Omicron BA.1 (32)	Non-Omicron (22)	NTC (10)
2-SARS-CoV-2 ORF1 assay	True positive (32) False negative (0)	True positive (32) False negative (0)	True positive (22) False negative (0)	True negative (10) False positive (0)
ERS31del assay	True positive (32) False negative (0)	True positive (32) False negative (0)	True negative (22) False positive (0)	True negative (10) False positive (0)
Ins214EPE assay	True negative (32) False positive (0)	True positive (32) False negative (0)	True negative (22) False positive (0)	True negative (10) False positive (0)

**Table 4 ijms-23-16153-t004:** Interpretation of the multiplex RT-qPCR assay results.

2-SARS-CoV-2 Assay	ERS31del Assay	Ins214EPE Assay	RP Assay	
FAM	HEX	ROX	Cy5	
+	−	−	+	SARS-CoV-2, non-Omicron
+	+	−	+	SARS-CoV-2 Omicron lineage, non-BA.1
+	+	+	+	SARS-CoV-2 Omicron lineage, BA.1
−	−	−	+	No SARS-CoV-2 RNA in the sample
−	−	−	−	No RNA in the reaction/ RNA extraction failed

**Table 5 ijms-23-16153-t005:** Primer and probe sequences for the multiplex RT-qPCR assay for the Omicron lineage detection with BA.1 determination.

Target	Name	Sequence
ORF1	2-SARS-CoV-2-ORF-1-F	AGAGCTATGAATTGCAGAC
ORF1	2-SARS-CoV-2-ORF-1-R	GGGAAATACAAAATTTGGACA
ORF1	2-SARS-CoV-2-ORF-1-P	FAM-AATTGGCAAAGAAATTTGACACCTTCA-BHQ1
ERS31del	N31-33del F	GTTTGGTGGACCCTCAGATT
ERS31del	N31-33del R	CAAGACGCAGTATTATTGGGTAAAC
ERS31del	N31-33del P	HEX-AGTAACCAGAATGGTGGGGCGCG-BHQ1
Ins214EPE	Ins214EPE F	ATATTCTAAGCACACGCCTATT
Ins214EPE	Ins214EPE R	GGCAAATCTACCAATGGTTCTA
Ins214EPE	Ins214EPE P	ROX-TGCGTGAGCCAGAAGATCTCCCT-BHQ2
Human RP	RP-CB-F	AGATTTGGACCTGCGAGCG
Human RP	RP-CB-R	GAGCGGCTGTCTCCACAAGT
Human RP	RP-CB-P	Cy5-TTCTGACCTGAAGGCTCTGCGCG-BHQ2

**Table 6 ijms-23-16153-t006:** Oligonucleotides’ premixing protocol for 100 reactions.

Reagents	μL	Concentration	Quantity
2-SARS-CoV-2-ORF-1-F	7.5	100 pmol/μL	300 nM
2-SARS-CoV-2-ORF-1-R	7.5	100 pmol/μL	300 nM
2-SARS-CoV-2-ORF-1-P	5	100 pmol/μL	200 nM
N31-33del F	2.5	100 pmol/μL	100 nM
N31-33del R	2.5	100 pmol/μL	100 nM
N31-33del P	1.25	100 pmol/μL	50 nM
Ins214EPE F	10	100 pmol/μL	400 nM
Ins214EPE R	10	100 pmol/μL	400 nM
Ins214EPE P	5	100 pmol/μL	200 nM
RP F	5	100 pmol/μL	200 nM
RP R	5	100 pmol/μL	200 nM
RP P	5	100 pmol/μL	200 nM
water	53.75		
Total volume	120		

## Data Availability

Sequences of all used genomes were deposited to GISAID, protocols shared on protocols.io: Ins214EPE assay protocol https://www.protocols.io/view/one-step-rt-pcr-ins214epe-assay-for-omicron-b-1-1-3byl4b2e8vo5/v1 (accessed on 18 November 2022). Omicron SARS-CoV-2 lineage and its BA.1 variant multiplex RT-qPCR https://www.protocols.io/view/sars-cov-2-omicron-detection-rt-qpcr-assay-with-b-kqdg3p9nql25/v1 (accessed on 18 November 2022). Other data will be provided upon reasonable request.

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
