# Peer review of "Detection of the Omicron SARS-CoV-2 Lineage and Its BA.1 Variant with Multiplex RT-qPCR"

_ijms, 2022, doi:10.3390/ijms232416153_

Round 1

Reviewer 1 Report

The authors produced a well-conducted study reporting significant contents on a novel time- and costs-sparing multiplex RT-qPCR for the simultaneous detection of both SARS-CoV-2 and different viral Omicron subvariants. It is a relatively new method, especially considering its use to identify the novel Coronavirus.

My impression of the manuscript is overall favorable: the study’s design is appropriate, the methods are suitable, the results look well-presented, and the relevance of the topic is very high. 

However, I suggest expanding the discussion section; this is the only part of the manuscript that appears poor in its contents. I’d like to show the reader what is the strength of the study and its limitations. Moreover, a comparison of the results with the findings that have been previously documented by other Authors would be appreciated. 

In addition, a full-text review by a native English speaker is mandatory for consideration for publication.

Author Response

The authors produced a well-conducted study reporting significant contents on a novel time- and costs-sparing multiplex RT-qPCR for the simultaneous detection of both SARS-CoV-2 and different viral Omicron subvariants. It is a relatively new method, especially considering its use to identify the novel Coronavirus.

My impression of the manuscript is overall favorable: the study’s design is appropriate, the methods are suitable, the results look well-presented, and the relevance of the topic is very high. 

Author response: Thank you!

However, I suggest expanding the discussion section; this is the only part of the manuscript that appears poor in its contents. I’d like to show the reader what is the strength of the study and its limitations. Moreover, a comparison of the results with the findings that have been previously documented by other Authors would be appreciated. 

Author response: Thank you for pointing this out. We have expanded the discussion section with a brief review of previously published papers on Omicron variant identification as well as with a description of the limitations of different methods.

In addition, a full-text review by a native English speaker is mandatory for consideration for publication.

Author response: English was additionally checked a native English speaker

Reviewer 2 Report

A multiplex test for detecting Omicron variants is a valuable contribution to current clinical practice. This paper goes a long way to showing how the authors' group has accomplished this. I found it very interesting.

Serious Issue - Statistics: Except for your probit analysis (figure 5), you have not included statistical evaluations of how sensitive or specific your test is at identifying cases correctly. What is the probability that you could have achieved the same results strictly by chance? If this is something you are leaving for a future, larger clinical trial of the test, you should state that. At this stage, the readers want to know with confusion matrices and the associated statistics that show how this test performs.

Discussion of Binding Schemes: In your section 2.1 on primer and probe design, I would like to know more about why the forward and reverse primers were chosen at the locations you show and how the probe fits within the binding scheme. This is especially an issue for Figure 1. The sequences you show could also be improved by showing a non-ambiguous location in the genome for the start and end points. In Figure 1, you should use the same graphic scheme as in 2 and 4, that is, putting the 3´ and 5´ indications on the end of the lines. 

Limit of Detection: You seem to be using LOD as a stand-in for sensitivity. It is certainly a component or precursor of sensitivity, but sensitivity is really (as I mentioned above) how many samples that are actually positive were detected as positive. Do you have that data yet?

Specificity: I like the way you have applied the test to samples with other respiratory viruses. Table 6 should show the 0 values rather than just leave spaces. It will take a reader a minute to understand what a blank cell in the table means.

Specific Issue with Fig 6: From W12-2022 on, the counts are quite small and the bars are hard to interpret. Put numeric labels in the graph to give an indication of what the counts actually were. This is also a place where some statistical hypothesis testing would help us understand what is a real trend and what is simply random noise.

In your discussion, you should give some consideration to how your test and its underlying technology can adapt to new variants, not necessarily sub-variants of omicron. We still have lots of letters in the Greek alphabet to go.

The paper is well written.

Author Response

Response to Reviewer 2 Comments

Point 1. A multiplex test for detecting Omicron variants is a valuable contribution to current clinical practice. This paper goes a long way to showing how the authors' group has accomplished this. I found it very interesting.

Response 1: Thank you!

Point 2. Serious Issue - Statistics: Except for your probit analysis (figure 5), you have not included statistical evaluations of how sensitive or specific your test is at identifying cases correctly. What is the probability that you could have achieved the same results strictly by chance? If this is something you are leaving for a future, larger clinical trial of the test, you should state that. At this stage, the readers want to know with confusion matrices and the associated statistics that show how this test performs.

Limit of Detection: You seem to be using LOD as a stand-in for sensitivity. It is certainly a component or precursor of sensitivity, but sensitivity is really (as I mentioned above) how many samples that are actually positive were detected as positive. Do you have that data yet?

Response 2: We agree with the reviewer’s assessment. Despite being sure of the high specificity and sensitivity of the developed assay, we agree that we provided not enough data to support this in the manuscript. Thus, we carried out one more experiment to assess specificity. We tested 96 samples and presented the results as a confusion matrix in the revised manuscript.

Point 3. Discussion of Binding Schemes: In your section 2.1 on primer and probe design, I would like to know more about why the forward and reverse primers were chosen at the locations you show and how the probe fits within the binding scheme. This is especially an issue for Figure 1. The sequences you show could also be improved by showing a non-ambiguous location in the genome for the start and end points. In Figure 1, you should use the same graphic scheme as in 2 and 4, that is, putting the 3´ and 5´ indications on the end of the lines. 

Response 3. Thank you for pointing this out. We unified all the binding schemes and added the viral genome coordinates. We added notes about primer and probe design as well.

Point 4. Specificity: I like the way you have applied the test to samples with other respiratory viruses. Table 6 should show the 0 values rather than just leave spaces. It will take a reader a minute to understand what a blank cell in the table means.

Response 4: Thank you for pointing this out. We filled out the spaces with the Nd abbreviation.

Specific Issue with Fig 6: From W12-2022 on, the counts are quite small and the bars are hard to interpret. Put numeric labels in the graph to give an indication of what the counts actually were. This is also a place where some statistical hypothesis testing would help us understand what is a real trend and what is simply random noise.

Response 5: Thank you for this suggestion. The figure was harmonized with the reviewer’s proposal.

In your discussion, you should give some consideration to how your test and its underlying technology can adapt to new variants, not necessarily sub-variants of omicron. We still have lots of letters in the Greek alphabet to go.

The paper is well written.

Response 6: Thank you! Note was added to discussion